# *Neurofibromatosis Type 1* Gene Alterations Define Specific Features of a Subset of Glioblastomas

**DOI:** 10.3390/ijms23010352

**Published:** 2021-12-29

**Authors:** Maximilian Scheer, Sandra Leisz, Eberhard Sorge, Olha Storozhuk, Julian Prell, Ivy Ho, Anja Harder

**Affiliations:** 1Department of Neurosurgery, Medical Faculty, Martin Luther University Halle-Wittenberg, Ernst-Grube-Straße 40, 06120 Halle, Germany; maximilian.scheer@uk-halle.de (M.S.); sandra.leisz@uk-halle.de (S.L.); julian.prell@uk-halle.de (J.P.); 2Department of Neuropathology, Institute of Pathology, Medical Faculty, Martin Luther University Halle-Wittenberg, Magdeburger Str. 14, 06112 Halle, Germany; eberhard.Sorge@uk-halle.de (E.S.); olha.storozhuk@uk-halle.de (O.S.); 3Department of Research, National Neuroscience Institute, Singapore 308433, Singapore; ivy_aw_ho@nni.com.sg; 4Institute of Neuropathology, University Hospital Münster, 48149 Münster, Germany; 5Brandenburg Medical School Theodor Fontane, Faculty of Health Sciences, Joint Faculty of the Brandenburg University, 16816 Neuruppin, Germany

**Keywords:** glioblastoma, neurofibromatosis, NF1, neurofibromin, mesenchymal, invasiveness, LRD domain

## Abstract

*Neurofibromatosis type 1* (*NF1*) gene mutations or alterations occur within neurofibromatosis type 1 as well as in many different malignant tumours on the somatic level. In glioblastoma, *NF1* loss of function plays a major role in inducing the mesenchymal (MES) subtype and, therefore defining the most aggressive glioblastoma. This is associated with an immune signature and mediated via the NF1–MAPK–FOSL1 axis. Specifically, increased invasion seems to be regulated via mutations in the leucine-rich domain (LRD) of the *NF1* gene product neurofibromin. Novel targets for therapy may arise from neurofibromin deficiency-associated cellular mechanisms that are summarised in this review.

## 1. Molecular Subtypes of Glioblastomas

Glioblastoma, previously denominated glioblastoma multiforme (GBM), represents the most common malignant glial primary brain tumour [1]. The brain tumour classification of the 4th edition of the World Health Organization (WHO) distinguishes between four glioma grades. Among these, GBM belongs to grade 4, which is the most aggressive type [2]. Although the survival of GBM patients improved significantly during the last few decades, median survival of approximately 15 months is still poor [3,4]. There is little but growing knowledge on risk factors for GBM such as ionising radiation or hereditary cancer syndromes. As currently assessed, only a small subset of GBM is associated with hereditary syndromes, such as neurofibromatosis type 1 (NF1), Lynch syndrome, or Li–Fraumeni syndrome [5]. 

The clinical course of GBM is diverse and depends strongly on tumour localisation. Signs due to raised intracranial pressure, epileptic seizures, and focal neurological deficits are typical [5]. The combination of surgery, radiation, and chemotherapy with temozolomide (TMZ) represents the current standard therapy [6,7]. Surgical radicality is of paramount importance for the overall outcome. Resection of >95% of tumour volume as demonstrated by contrast-enhanced Magnetic Resonance Imaging (MRI) (“gross total resection”) has been shown to improve overall and progression-free survival significantly [8]. Nevertheless, gross total resection is not always achievable, even with integrating intraoperative imaging or 5-aminolevulinic acid fluorescence. Eloquent tumour localisation close to cortical or subcortical structures of major functional relevance such as the primary motor cortex or the deep tracts of the language system in the dominant hemisphere limit the extent of resection. Incomplete resection has been demonstrated to influence treatment outcome and negatively affect patient’s survival [9]. Recently, novel therapeutic approaches such as a combination of lomustine with TMZ as well as the use of Tumour-Treating Fields (TTFields) [4,10] are associated with improved survival. The effectiveness of these and possible future modalities is influenced by GBM molecular subclasses. Therefore, the subclasses are of high clinical importance and value.

GBM shows a diverse histological pattern that consists of necrosis, microvascular proliferation, increased mitotic activity, anaplasia, and invasion. Among these features, necrosis and microvascular proliferation conventionally distinguish GBM from high-grade astrocytoma [11]. In the last couple of years, the impact of molecular features on diagnosis increased dramatically. The combination of histological and molecular characteristics into an integrated diagnosis has become indispensable for appropriate diagnostic procedures and therapeutic planning [12,13]. For this purpose, next-generation-based genomic profiling entered clinical practice. Using specifically methylation profiling, GBM can be divided into six methylation subgroups [14]. For instance, gliomas that occur in NF1 patients are assigned to LGm6, which is a poorly defined methylation class subgroup [15].

At least two molecular markers have been established for a minimal clinical routine. Methylation of the *O6-methylguanine-DNA methyltransferase* (*MGMT*) promoter still serves as an important predictive marker and is associated with a better response to chemotherapy [6,7]. Additionally, hotspot mutations of the *isocitrate dehydrogenase (IDH*) gene that occur early in gliomagenesis are important diagnostic and prognostic markers in glioma subtypes [5]. GBM was classified into primary GBM, *IDH-*wild-type, and secondary GBM. According to the most recent 5th edition of the WHO classification diffuse astrocytomas are classified as *IDH*-wild-type GBM WHO grade 4 and *IDH-*mutant astrocytomas WHO grade 2, 3, or 4. Including now specific genetic events into diagnostics, microvascular proliferation or necrosis or one of the following genetic alterations such as *telomerase reverse transcriptase (TERT*) promoter mutation, *epidermal growth factor receptor (EGFR*) gene amplification, and +7/−10 chromosome copy number changes are sufficient to diagnose GBM [16,17]. 

Based on gene expression and genomic clustering, The Cancer Genome Atlas (TCGA) project established four GBM subclasses: classic (Cl), neural (N), proneural (PN), and mesenchymal (MES) [18]. Since recent studies did not identify the neural subtype securely, its existence is controversially discussed. Some authors claim contamination through normal neural tissue as an explanation [14,19,20] (Figure 1).

A high expression of angiogenesis or proliferation-associated genes, as well as poor median overall survival (about 14.7 months), is characteristic for the Cl subtype [5,21,22]. Particularly, molecular events comprise *EGFR* amplification or mutations and focal 9p21.3 homozygous deletions, including the *cyclin-dependent kinase inhibitor 2A (CDKN2A*) gene [18,21,23]. Amplification of chromosome 7 (+7), a loss of chromosome 10 (−10), and *TERT* promoter mutations, which are considered a characteristic event in GBM, are typically present in the Cl subtype. Additionally, loss of tumour protein 53 (TP53) and mutation in the *Phosphatase and Tensin homolog (PTEN*) gene is frequently observed in the Cl subtype [18,22]. The DNA methylation subtype “receptor tyrosine kinase (RTK) II” corresponds to the Cl subtype [24]. In contrast, the PN subtype corresponds to the RTK I methylation subtype. Patients with PN GBM subtype show a more favourable median survival time of approximately 17 months [5,22]. In contrast to the Cl subtype, there is a lack of *PTEN* and *EGFR* mutations. A specific alteration in the PN subtype is focal amplification of the 4q12 locus harbouring the *platelet-derived growth factor receptor A (PDGFRA*) gene [18,21]. In addition, the PN subtype can be found among different types of gliomas (WHO grade 2 and 3) and is often associated with *IDH* mutations [20,25]. *TP53* mutations and loss of heterozygosity are frequent events, whereas chromosome 7 amplification paired with chromosome 10 loss is distinctly less prevalent [18,19]. The MES GBM subtype is considered most aggressive and is associated with the worst median overall survival of 11.5 months. MES tumours express mesenchymal markers such as chitinin-3-like protein (CHI3L/YKL40) and vimentin, and they downregulate proneural markers such as oligodendrocyte transcription factor 2 (Olig2), thus showing an upregulation of angiogenesis and proliferation-related genes [14,18,22]. The poor outcome of MES GBM patients that we similarly experience in our clinic (Figure 2) challenges research to identify specific therapies for this subtype. 

One of the hallmarks of GBM is intra-tumour heterogeneity [5,26]. Recent studies demonstrated the presence of different GBM subtypes and progenitor cells in the same tumour [19,26,27]. Even in primary cell culture, this phenomenon was observed [28]. The shift from the PN to MES subtype is known as proneural–mesenchymal transition (PMT) and is characterised by an increased malignant behaviour [26,27]. PMT is associated with the downregulation of E-cadherin and the upregulation of N-cadherin, vimentin, and fibronectin [26]. Principally, the MES phenotype is a result of alternative processes, including intrinsic processes due to mutations and changes of the tumour microenvironment as well as extrinsic factors due to treatment [29]. Chemokines and cytokines secreted by other cellular components of the tumour microenvironment, as well as reactive oxygen species produced because of radiation and chemotherapy, were also shown to induce mesenchymal subtype transition. The recruitment of macrophages, stem cells, progenitor cells, the *NF1* mutation, the cell of origin, and localisation and therapeutic effects due to chemo-, radio-, and antiangiogenic therapy are supposed to result in a mesenchymal transition [29]. Poor response to radiotherapy is associated with CD44 expression and NF-κB activation [30]. Important regulators of the proneural to mesenchymal transition in GBM are tumour-associated (M2) macrophages that produce growth factors and promote tumour growth and proliferation as well as neutrophils [19]. Tumour-infiltrating lymphocytes are enriched in MES GBM and are strongly associated with *NF1* mutations [19,26,27,31]. 

## 2. The *Neurofibromatosis Type 1* (*NF1*) Gene in Normal Tissue

The *NF1* genomic DNA sequence is mapped on chromosome 17q11.2 [32,33], and its protein, neurofibromin, spans over a large size of 280 kb [34]. The *NF1* gene comprises 57 constitutive and at least three alternatives spliced exons. *NF1* pseudogenes (on chromosomes 2, 12, 14, 15, 18, 21 and 22) may complicate molecular diagnosis [35,36]. 

Many studies indicate the importance of *NF1* splice variants, of which five are analysed on an experimental level [37,38,39,40]. In general, the gene product neurofibromin isoform type 2 (NP_000258.1, 2818 amino acids (aa)) is expressed ubiquitously and shows a 10 times higher Rat sarcoma GTPase activating protein (Ras-GAP) activity than isoform 1 (NP_001035957.1; 2839aa) [41]. It is preferentially expressed in differentiated cells [37,42]. Isoform 1 contains 21 additional amino acids encoding for the alternatively spliced exon 23a. The alternatively spliced exon 23a (exon 31 according to the new nomenclature) [43,44] is placed amid the GTPase-activating domain (GAP) related domain (GRD). Therefore, the Ras-GAP activity depends on 23a exon splicing. Isoform 1 represents the most abundant isoform [45] and is expressed in adult tissues of neural crest lineage [46]. Still, there is evidence for a tissue-specific accumulation of splice variants, the co-existence of different splice variants in the same cell type, and a correlation between the protein expression level and tissue type [39,47,48]. It was also shown that benign tumours and peripheral nerves share the same spliced RNA expression profile, indicating that in benign tumours, *NF1* may be spliced identically. In the CNS, *NF1* isoform 2 is preferentially expressed in pure glial cultures, while isoform 1 is predominantly expressed in neuronal cells [49]. Among the different splice variants, the National Center for Biotechnology Information (NCBI) reference sequence NM_000267.3 is most widely used for variant analysis. The accumulation and expression of splice variants are specific to developmental stage and tissue [50]: the splice variant resulting from alternative splicing of exon 9a adds ten amino acids to the protein sequence and is mainly located in the central nervous system. Studies in mice showed increased expression levels during the first postnatal week, suggesting a role for the maturation and differentiation of neurons [39,51,52]. The alternative spliced exon 10a-2 is located between exon 10a and 10b and adds fifteen additional amino acids. The resulting additional motive forms a transmembrane segment that does not appear in other variants. Although expression was detected in every human tissue, pointing to a housekeeping function [40]. Alternative splicing of exon 48a results in additional eighteen amino acids and is discussed to play a role in the differentiation of foetal and adult cardiac and skeletal muscle [38,53,54]. Interestingly, alternative spliced exons 29 and 30 lead to three different protein isoforms: ex29-, ex30-, and ex29-/30- [55]. Except for ex29-, which is only apparent in the brain, these variants are ubiquitously expressed, but no variant-specific function has been described so far.

Structural and functional analysis of neurofibromin (Protein Data Bank P213599) revealed a complex domain architecture (Figure 3) [50]. While the precise role for many domains is still not fully understood, the GRD is well characterized. GRD promotes the hydrolysis of active Ras-GTP to the biologically inactive form of Ras-GDP [56], thereby negatively regulating the Ras/mitogen-activated protein kinase (Ras/MAPK) pathway. Interestingly, neurofibromin forms a high-affinity homodimer [57]. Mutant variants may dimerize with functional wild-type neurofibromin. A dysfunctional complex might be a target for proteasomal degradation and inhibit tumour-suppressor activity. Whether this plays a role in disease development or correlates with NF1 phenotypes remains to be investigated. 

Upstream of neurofibromin, mainly transmembrane receptor tyrosine kinases (RTKs) regulate extracellular ligand binding and transduce signals into the cells. They regulate signalling cascades such as the RAS/ERK pathway and therefore interfere with neurofibromin. Therapies using RTK inhibitors may fail when *NF1* mutations abrogate the effect on the cascade. Interestingly, *Anaplastic Lymphoma Kinase* (*Alk*) was shown to co-localise and interact with neurofibromin in Drosophila and was demonstrated to activate neurofibromin-regulated RAS signalling in the nervous system [70]. A direct interacting partner of neurofibromin is also the membrane-bound late endosomal/lysosomal adaptor and MAPK and mTOR1 activator (LAMTOR), which is a negative regulator of the mTOR pathway [71,72]. Although other interacting partners are very important, such as Sprouty-related and EVH1 domain-containing protein 1 (SPRED1), which recruits neurofibromin from the cytosol to facilitate the transport to the plasma membrane, they will not be discussed in detail here. 

## 3. The *Neurofibromatosis Type 1* (*NF1*) Gene in Neoplasia

Due to its large size and complexity, *NF1* is one of the most frequently mutated genes in men and in cancers [73]. The Human Gene Mutation Database (HGMD^®^ Professional 2021.2) currently lists 3084 *NF1* germline mutations, and TCGA reports 1110 somatic mutations. The majority of these mutations lead to truncated neurofibromin, and about 30% of mutations lead to altered splicing [74]. Despite the high number of mutations, there are few mutation hotspots. NF1 patients with gliomas do not show the involvement of specific *NF1* gene regions [15]. In fact, very few genotype–phenotype correlations exist, except for ahigher and more aggressive tumour load in patients with microdeletions [75]. 

Neurofibromin regulates cell growth and survival through several downstream signalling effectors such as Ak strain transforming/protein kinase B (Akt), mammalian target of rapamycin (mTor), and protein kinase A (PKA) by accelerating the conversion of Ras hydrolysis via the catalytic central GRD [50,69]. Some of the Ras-induced proteins are involved in EMT and have been shown to be increasingly expressed in *NF1*-deficient malignant peripheral nerve sheath tumours (MPNST) [76,77]. Neurofibromin deficiency promotes not only EMT but also resistance to inhibitors along the MAPK pathway [78,79]. Therefore, in cancer, *NF1* mutations act not only as drivers but contribute to therapy resistance [69]. *B rat fibrosarcoma*
*(BRAF)* mutations, upregulation of *EGFR*, or activation of mitogen-activated protein kinase (MEK) are associated with resistance as reported for melanoma, neuroblastoma, lung cancer, and other lesions [80,81,82,83]. Loss of *NF1* also activates cell motility by negative regulation of the Rho/Rho-associated coiled-coil-containing protein kinase (ROCK)/LIM domain kinase (LIMK), cofilin pathway, which induces the dynamic reorganisation and turnover of actin filaments [77,84]. Consequences of neurofibromin deficiency in tumours are schematically summarised in Figure 4.

Patients with autosomal dominantly inherited NF1 are prone to develop benign peripheral nerve tumours known as neurofibromas, which is the hallmark of the disease. Cutaneous neurofibromas arise due to mutations of both copies of the *NF1* tumour suppressor gene in Schwann cells (biallelic inactivation). It is important to point out that only the Schwann cells are *NF1 -/-*, while other components within the microenvironment are *NF1 +/-* [85]. The development of NF1 and the subsequent reprogramming of Schwann cells have been extensively reviewed and are not the focus of this review [75,86,87,88,89,90,91]. Other NF1-associated tumours comprise plexiform neurofibromas (30–50%), optic pathway gliomas (15–20%), MPNST (10–15%), and others [73,92]. Recently, diagnostic criteria have been updated [93]. The mutational spectrum includes missense or nonsense mutations (33%), small deletions (26%), splicing substitutions (15%), small insertions/duplications (11%), and gross deletions (over 20 bp, 11%). About half of all NF1 patients display new mutations. In principle, NF1-associated benign lesions in NF1 patients acquire a somatic *NF1* loss of heterozygosity (LOH) to be initiated, which accompanies the germline *NF1* mutation. For the development of pre-malignant and malignant lesions, additional genetic hits are necessary. This genetically defined increased risk of NF1 patients to develop malignancies from their benign lesions still reduces life expectancy in NF1 [75]. 

In contrast, lesions that are independent of NF1 can develop when somatic *NF1* loss of heterozygosity occurs. These lesions include not only GBM but breast cancer, uterine cancer, and melanoma, among others [94,95]. In these cancer types, *NF1* is co-mutated with other tumour-suppressor genes such as *p53*, *PTEN,* and *BRAF*, and others. The frequency of mutations and copy number variation loss events of *NF1* in different tumour entities is variable (Table 1). Attempts have been made to delineate the genotype–phenotype correlation of these mutations. 

## 4. The *Neurofibromatosis Type 1* (*NF1*) Gene in GBM

*NF1* is mutated in approximately 13–14% of GBM patients according to the TCGA PanCancer Atlas GBM database [94,95]. Most (78%) of the pathogenic variants are generated by frameshifts, single nucleotide polymorphisms (SNP), or splice variants, resulting in truncation of the full-length neurofibromin and nonsense-mediated ribonucleic acid (RNA) decay. Although GBM shows an increased incidence of *NF1* mutation, *PTEN* (35%), *TTN* (33%), *TP53* (32%), *EGFR* (27%), *FLG* (20%), and *MUC16* (18%) display higher mutation rates (Figure 5A). Patients with *NF1*-mutated GBM have a lower overall survival than those patients without (Figure 5C). Interestingly, 53% of the mesenchymal GBM subtype are *NF1* mutated [15]. In an *Nf1*+/− mouse model, loss of *NF1* function was shown to increase astrocyte proliferation [43]. In astrocytes, loss of neurofibromin causes the selective hyperactivation of KRAS rather than HRAS [44]. 

## 5. Mesenchymal Glioblastomas Accumulate *Neurofibromatosis Type 1 (NF1) Gene* Alterations

The MES molecular subtype accounts for approximately 35% of all adult high-grade gliomas [18,21,23] and is primarily characterised by the loss or deregulated expression of *NF1* [19,30,96,97]. The MES subtype is also associated with mutations of *TP53* and *RB1* [22], enhanced activity of the tumour necrosis factor superfamily (TNF), and nuclear factor kappa-light-chain-enhancer of activated B cells (NF-κB) pathways with co-mutation of *PTEN* [18,20,21,94,95]. The upregulation of these genes is accompanied by higher overall necrosis and inflammatory infiltrates. 

It is known that *NF1* mutations correlate with high levels of leukocytes in different tumour types [98]. *NF1* mutations lead to altered levels of cytokines, mast cells, macrophages, microglia, T and B cells, and they both directly affect immune cells and indirectly affect interactions between different *NF1*-mutated cells important for the tumour microenvironment [99,100,101]. In NF1-associated neurofibromas and MPNST, which are derived from peripheral glia, up to 30% of cells are macrophages. This finding led to the current hypothesis of neurofibroma formation in NF1: tumour initiation due to *NF1* loss is followed by macrophage and mast cell recruitment, which is then followed by the recruitment of T and dendritic cells to enable tumour formation [102]. Half of NF1-associated low-grade gliomas were detected to harbour an immune signature, infiltrates of T cells, and increased neoantigens [15]. Therefore, the role of *NF1* loss for microenvironment and tumour formation may well be adapted to the central nervous system-derived malignant glial tumours, the GBM, although the literature is sparse [99]. In a recent animal model, midkine being produced by *NF1* mutant neurons activates T lymphocytes and maintains glioma growth [103].

NF1-related tumours are associated with the abnormal secretion of chemokines such as C-C motif ligand (CCL) 15, CCL 2, and macrophage colony-stimulating factor (M-CSF), leading to an increase in tumour-associated macrophages (TAM) and microglia [99,104]. Especially in GBM, loss of neurofibromin is clearly associated with the attraction of macrophages (tumour-associated macrophages, TAM) or microglia [19]. Immunotherapy strategies targeting TAM have certain potential but have only been studied in mouse models and small clinical trials. CCL antibodies or M-CSF receptor inhibitors reduced glioma cell invasion and resulted in longer overall survival in glioblastoma mouse models [105,106,107]. In addition, the activation of immune cell response with immune checkpoint inhibitors and cytokine therapy (IL-2, IFN-ß) leads to prolonged patient survival [108,109,110,111,112]. Thus, numerous ongoing clinical trials are investigating the effect of PD-1/PD-L1 antibodies in glioma. Moreover, immunotherapies seem to be not only a promising strategy for mesenchymal gliomas, but they are also an important treatment option for NF1-related melanomas, lung carcinomas, or MPNST. Recently, loss of *NF1* was shown to modulate *FOS like 1, AP-1 transcription factor subunit (FOSL1)* expression, which is a key regulator for stemness, mesenchymal shift, and plasticity [113]. Transcription factor FOSL1 is overexpressed in cancer and associated with worse outcomes and EMT as well as with glioma malignancy [113]. The authors demonstrated that *FOSL1* depletion in *NF1* mutant human brain tumour stem cells and *KRAS* mutant mouse neural stem cells resulted in the loss of the MES signature and a reduction in stem cell properties. They first proved that *NF1* mutations act via the NF1–MAPK–FOSL1 axis in MES gliomas as they increase *FOSL1* RNA and protein expression and therefore activate the expression of the MES gene signature and inhibit the non-MES gene signature [113]. 

The important role of *NF1* to regulate FOSL1 expression explains the proneural to mesenchymal transition in tumours that acquire *NF1* mutations such as MES GBM (Figure 6). This interaction otherwise hints to novel therapeutics against the FOSL1 axis, the immune system, and combined approaches against several cellular components. 

## 6. *NF1* Mutations and Glioma Invasiveness

Neurofibromin regulates the dynamic reorganisation and turnover of actin filaments through its interacting partners such as Ras-related C3 botulinum toxic substrate 1 (Rac1) [114,115], Lim kinases (LIMK1/2) [77,84,114,116], syndecan-2 [117], and focal adhesion kinase (FAK) [118,119,120] among others. Neurofibromin binding to syndecan-2 induces actin polymerisation and filopodia formation in dendrites [117]. Along the same line, neurofibromin interaction with FAK regulates cell migration [120]. Given its role as a modulator of cytoskeletal and focal adhesion as well as a negative regulator of RAS signalling, mutations, or loss of *NF1* results in disruption of the extracellular matrix and induction of EMT. Indeed, we and others have shown that the deregulation of neurofibromin signalling enhanced cancer cells invasion and migration [76,100,121,122,123] with an increase in EMT markers such as vimentin and Chitinase-3-like protein 1 (CHI3L/YKL40) expression. 

Our group recently showed that the leucine-rich domain (LRD, aa1558–1951, isoform 2) of neurofibromin, which consists of the Sec14-pleckstrin homology (PH) domain (aa 1558–1817) and part of the Heat-like repeat (HLR; aa1818–1951), inhibits *NF1*-loss induced cell invasion in human glioma stem cells (GSC) and orthotopic mouse glioma model independent of RAS [122] (Figure 7). Mutation screening performed on the TCGA PanCancer Atlas GBM database identified 10 mutations in the LRD of which three are located within the 1818–1951 HLR region (D1828N, W1931*, and R1947*) [94,95]. Unlike the wild-type (wt)-LRD that suppresses glioma cell invasion, the inhibitory effect is lost in both D1828N and W1931* pathogenic mutants. We further narrowed down the region critical for LRD function to a 42-aa peptide between 1818 and 1860. This 42-aa peptide suppresses glioma invasion to levels significantly lower than that of wt-LRD, suggesting a critical role of the 1818–1860 region in regulating glioma cell motility. It is not clear how this peptide mediates its function. The peptide may interact with protein(s) that regulate cell motility and ECM remodelling, since one of the roles of the HLR is protein–protein interaction. This hypothesis is consistent with Welti’s and Scheffzek’s findings that the Sec14-PH domain of LRD interacts with phospholipids for membrane localisation [68,124]. The D1828X and W1931X mutations are detected in patients with cutaneous melanoma, colon carcinoma, diffuse large B cells lymphoma [125], and infiltrative breast carcinoma (cBioportal TCGA database, Tumour suppressor gene database, NCBI dbSNP, ClinVar, and Human Proteome Variation Database). Additionally, the mutation W1931X nonsense variant has been previously reported to be associated with NF1 [126,127,128]. 

Other domains involved in cell invasion and migration include the GRD, Sec14-PH domain, pre-GRD NF11-1163, and the CTD. Both GRD and Sec14-PH domains mediate cell migration and invasion through LIMK2, which is a kinase in the Rho/ROCK/LIMK2/cofilin pathway. The overexpression of GRD has been shown to alter cellular morphology to inhibit cell invasion via LIMK2 dephosphorylation of cofilin [84]. Similarly, the interaction between LIMK2 and the Sec14-PH domain prevents the activation of LIMK2 by ROCK due to steric hindrance, thus resulting in actin depolymerisation via cofilin [77]. Interestingly, the Sec14-PH domain interacts with LIMK2 exclusively and does not bind to LIMK1. By contrast, the pre-GRD NF11-1163 domain does not bind to ROCK and Ras. Rather, it negatively regulates the Rac1/p21 Rac-activated kinase (Pak)1/LIMK1/cofilin pathway [114]. By inducing the depolymerisation of cofilin, the NF11-1163 that contains the cysteine-serine-rich domain (CSRD) inhibits cell migration and invasion. Neurofibromin also binds to the N-terminal of FAK [118] and syndecan-2 via the CTD domain. Whether mutations in these domains will affect the depolymerisation of cofilin is unknown; hence, cell invasion is unsolved. Since most of the alterations generate truncation mutants, it is conceivable that mutations observed in glioma patients will most likely abolish the interaction between the neurofibromin domains and their substrates and destabilise the actin filament organisation, thus affecting cell invasion. Of note, the NF11-1163 region is highly conserved. Mutations in this domain are found in a higher proportion of NF1 patients with optic pathway glioma [129,130]. Protein kinase C (PKC)-α phosphorylation on the serine residues within CSRD induces the association of neurofibromin with the actin cytoskeleton [60]. Thus, a mutation in the CSRD may affect the actin reorganisation. 

It is important to note that although mutations identified from the cBioportal database may help to dissect the functional significance of the neurofibromin domains in GBM, some of these mutations are different from the mutations observed in NF1 patients since neurofibromin is a macromolecule without any mutational hotspot. In addition, most studies were done using specific neurofibromin domains in the absence of the entire *NF1* gene; thus, they may not offer sufficient power to detect potential genotype–phenotype correlations.

## 7. Conclusions

Despite advances in surgery and molecular therapeutics, the prognosis for patients with GBM remains dismal. The highly infiltrative and heterogenous nature of the tumour is rendering standard therapeutic strategies ineffective. *NF1* is one of the driver genes for MES GBM. In this review, we discussed the molecular characteristics of MES GBM, *NF1* gene mutation, and dysregulation in NF1-associated and non-NF1 associated cancers, particularly GBM. However, many questions remain unanswered. MES GBM gene expression is influenced by dysregulated neurofibromin signalling and the tumour microenvironment [131]. In *NF1*-null or silenced MES GBM, the microenvironment is heterogenous with a hypoxic core and perivascular niche, each secreting different cytokines and chemokines that drive tumour malignancy. Given the complexity of the bi-directional interaction, the design of therapeutics must take into consideration the dynamic crosstalk among the various players such as glioma cells, immune cells (immunosuppressive versus pro-inflammatory), and endothelial cells, among others. Macrophages and microglia cells secrete factors that promote tumour growth. Are we able to re-educate these cells in the *NF1*-null microenvironment to achieve the anti-tumour function? Studies conducted by Pyonteck et al. using a brain-penetrant inhibitor of colony-stimulating factor 1 receptor (CSF-1R) showed a significant decrease in pro-tumourigenic tumour-associated macrophages [106], suggesting that blocking CSF-1R signalling may re-educate the immunosuppressive macrophage to pro-inflammatory cells. Another CSF-1R tyrosine kinase inhibitor, PLX3397, prevented the differentiation of monocytes into immunosuppressive macrophages [132]. Unfortunately, PLX3397 was ineffective in a phase II trial in treating recurrent GBM [107]. Thus, understanding the intricate relationship between these cells and their associated gene expression changes may help develop more effective immunotherapeutics. Given that GBM subtypes are not static, it is evident that multiprong therapy may afford a better therapeutic outcome. Previous publications have shown that CEBP-β, STAT3, NF-kB, and FOSL2 are some of the transcription factors (TFs) that play a role in *NF1*-loss-associated MES transition [133]. Among these TFs, STAT3, and CEBP-β have been shown to associate with the hypoxic microenvironment [29,134], which is enriched with immunosuppressive tumour-associated macrophages [135]. Gabrusiewicz et al. showed that GBM-derived exosomes triggered the release of STAT3 in monocytes and led to the upregulation of programmed death-ligand 1 (PD-L1) and a shift to the immunosuppressive phenotype [136]. Several STAT3 inhibitors are currently in clinical trials. These inhibitors were designed to be used concurrently with conventional radiation (NCT03514069) and chemotherapy (NCT02315534). Other inhibitors that target the molecules in the STAT3 pathway, such as JAK1/JAK2, are also being evaluated in phase I trial for patients with newly diagnosed GBM (NCT03514069). While we await the results from these trials, identifying other *NF1*-loss associated master regulators and their inhibitors may improve the treatment options for patients with MES GBM.

## Figures and Tables

**Figure 1 ijms-23-00352-f001:**
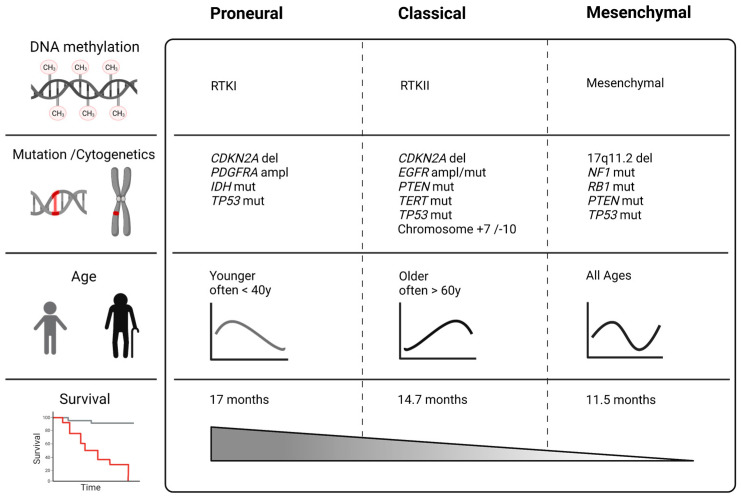
GBM subclasses based on The Cancer Genome Atlas (TCGA) project and Verhaak classification with the most prevalent genetic abnormalities [18]. Novel WHO classification and methylation profiling differentiate between more subgroups of malignant astrocytoma, but in comparison, the MES subtype is still associated with very poor survival. Created with BioRender (ED2349TVGA, 25.10.2021).

**Figure 2 ijms-23-00352-f002:**
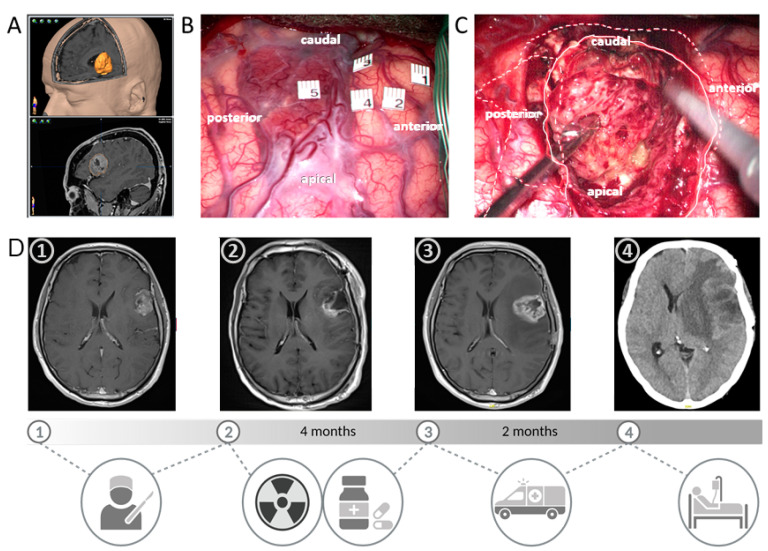
This case of a patient with a mesenchymal GBM highlights the unfavourable clinical course compared to other subtypes impressively. A 42-year-old male patient presenting with seizures demonstrated a contrast-enhancing lesion in Broca’s left-sided area on MRI (**A**,**D**) and bilateral activation by functional MRI. Awake craniotomy with cortical (**B**) and subcortical mapping was scheduled due to the highly eloquent localisation of the tumour, which was both clearly visible on the brain surface (**B**) and subcortically (**C**) with typical necrotic tissue and pathological vessels. Here, 86% of the contrast-enhancing lesion was resected (**D**, first to the second picture), as quantified by MRI volumetry, while a total resection was not feasible due to close eloquent subcortical pathways as demonstrated by bipolar electrical stimulation intraoperatively. Postoperatively, the patient suffered from dysphasia. Radiotherapy (60 Gy) and TMZ therapy were simultaneously administered and discontinued after radiotherapy by the patient himself. Only 4 months after surgery, the patient developed progressive dysphasia and hemiparesis (**D**, third picture). A second surgery was not considered to be feasible, and chemotherapy, according to Stupp’s protocol, was administered together with TTFields. However, the patient again presented an emergency admission with rapid clinical and radiological progression 2 months later (**D**, fourth picture). Lacking alternatives, palliative care was initiated. Created with BioRender (OI2349TKM9, 25 October 2021).

**Figure 3 ijms-23-00352-f003:**
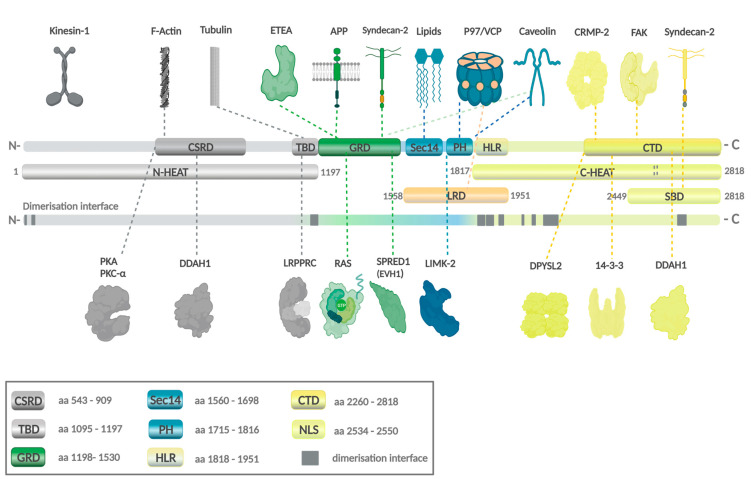
Domain architecture of neurofibromin: The lemniscate NF1 complex is formed by a head-to-tail dimer of an N-terminal HEAT domain (N-HEAT) and a C-terminal HEAT domain (C-HEAT) [58]. The GRD domain possesses a Ras-GAP function. The cysteine and serine-rich domain/Ras-GTPase activating protein domain (CSRD) and the C-terminal domain (CTD) harbour phosphorylation sites, and they are anticipated to regulate GAP-activity when phosphorylated by protein kinase C (PKC) and cAMP-dependent protein kinase A (PKA) [59]. Phosphorylation at the CSRD domain potentiates the Ras-GAP activity [60], while phosphorylation at multiple serine residues in the CTD prolongs activation of the Ras/extracellular signal-regulated kinase (ERK) pathway [61], mediates nuclear import of neurofibromin during the cell cycle [62], and facilitates neurofibromin interaction with 14-3-3 that negatively regulates the GAP-activity [63]. CTD includes a nuclear localisation signal (NLS). Aside from binding to Ras, the GRD also interacts with tubulin via its tubulin-binding domain (TBD) motive [64]. Bipartite module Sec14-homologous segment and pleckstrin homology (PH)-like domain binds phospholipids and is structurally well characterised [65]. Caveolin (CAV1) binding sites are spread over the GRD and the Sec14/PH-domain [66]. CAV1 interacts with Musashi-2 (MSI2), and knockdown of MSI2 elevates the CAV1 protein expression, inhibiting the ubiquitinylation of CAV1 [67]. The leucine-rich domain (LRD) consists of 393 amino acids and includes SEC-PH and Heat-like repeat (HLR) domains. It is involved in membrane localisation through the binding with lipids, actin remodelling through the Rho–ROCK pathway, and dendritic spine formation through VCP. As a neurofibromin creates a high-affinity dimer, on the bottom with the gray colour are shown primary dimerisation interfaces [58]. The figure was created with BioRender.com (XR2390NUCR, 27.11.2021) and adapted from [68,69].

**Figure 4 ijms-23-00352-f004:**
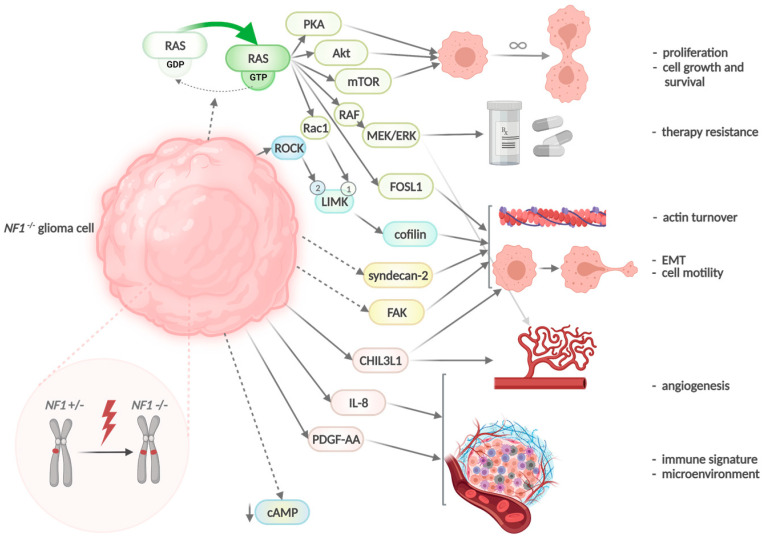
A consequence of *NF1* loss of function in cancer: *NF1* deficiency prevents inactivation of Ras through GTP hydrolysis and leads to upregulation of Ras signalling. It is associated with increased tumour proliferation, EMT, invasion, cell motility, and therapy resistance. Moreover, the hyperphosphorylation of LIMK1 in Ras- or LIMK2 in a ROCK-dependent manner results in the activation of cofilin pathways, which leads to changes in actin cytoskeleton and cell motility. The inability of the C-terminal domain of neurofibromin to bind to focal adhesion kinase and syndecan-2 induces cell detachment and might facilitate the epithelial to mesenchymal transition. This process is supported by the upregulation of FOSL1 and increased secretion of chitinase-3-like protein 1 (CHI3L1). Loss of *NF1* changes also the tumour microenvironment and angiogenesis by the enhanced secretion of platelet-derived growth factor AA (PDGF-AA) and interleukin-8 (IL-8). In patients with NF1, the somatic *NF1* hit accompanies a germline mutation (*NF1 -/-*), which corresponds to a complete loss of function of neurofibromin, a classical tumour suppressor. In non-NF1-associated *NF1*-altered GBM, the somatic *NF1* hit accompanies other primary genetic alterations that might act similar, e.g., affect the MAPK pathway. Created with BioRender (ZE2345L4TH, 24 October 2021).

**Figure 5 ijms-23-00352-f005:**
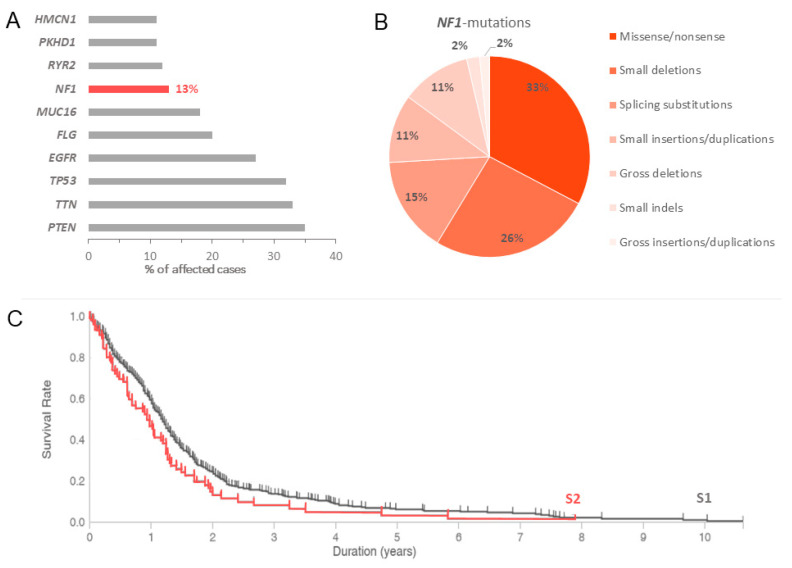
*NF1* mutations in glioblastoma cases (not associated with NF1). (**A**,**B**): Distribution of most frequently mutated genes in glioblastoma and spectrum of mutations. (**C**): Overall survival plot of patients with *NF1* mutated glioblastoma (S2; n = 79) compared with *NF1* wild-type cases (S1; n = 516). Log-Rank test *p*-value 0.017. Data were derived from the Genomic Data Commons Data Portal (TCGA database).

**Figure 6 ijms-23-00352-f006:**
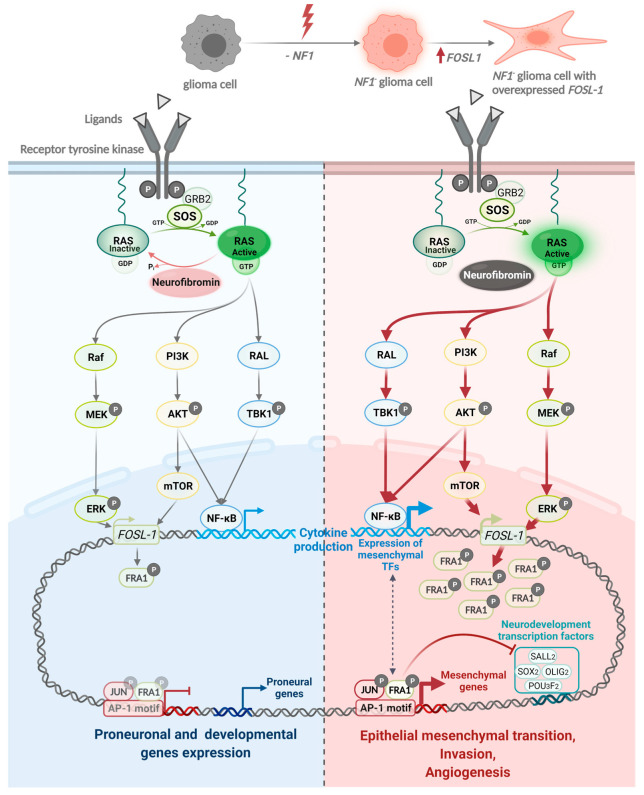
FOSL1 is regulated by neurofibromin and its role for proneural to mesenchymal transition. Created with BioRender.com (SY2390LZAW, 27 November 2021).

**Figure 7 ijms-23-00352-f007:**
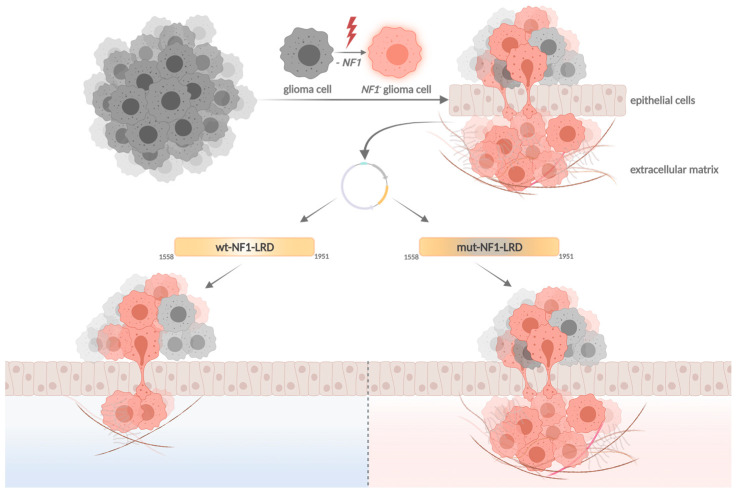
NF1 alteration of the leucine-rich domain (LRD) of neurofibromin promotes glioma invasiveness. Created with BioRender (UE233WMPOB, 23 October 2021).

**Table 1 ijms-23-00352-t001:** Loss of *NF1* in a spectrum of neoplasms underlines its tumour suppressor function (mutation frequencies of *NF1* according to the Genomic Data Commons Data Portal).

Tumour Entity	Frequency of Somatic Mutation	Frequency of CNV Loss Events
Uterine Corpus Endometrial Carcinoma	19.62%	5.69%
Melanoma	16.63%	3.63%
Glioblastoma multiforme	12.98%	3.04%
Lung Squamous Cell Carcinoma	12.73%	5.78%
Lung Adenocarcinoma	12.52%	3.31%
Angiosarcoma	11.11%	1.76%
Cervical Squamous Cell Carcinoma and Endocervical Adenocarcinoma	10.03%	2.78%
Adrenocortical Carcinoma	9.78%	3.33%
Stomach Adenocarcinoma	9.55%	1.16%
Paragangliomas and Glomus Tumours	9.50%	10.83%
Bladder Urothelial Carcinoma	8.98%	2.94%
Ovarian Serous Cystadenocarcinoma	7.57%	14.36%
Sarcoma	7.17%	17.31%
Breast Invasive Carcinoma	5.58%	6.06%

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
