# Peer review of "Neurofibromatosis Type 1 Gene Alterations Define Specific Features of a Subset of Glioblastomas"

_ijms, 2021, doi:10.3390/ijms23010352_

Round 1

Reviewer 1 Report

It would be helpful to add more detail regarding any data out there that suggests that GBM patients (or other cancer types) with NF1 mutations respond better to immunotherapy. This is implied, but giving clinical examples would make this more impactful (referring to lines 283-295)

I would also suggest adding a figure showing the role of FOSL1 both in terms of its regulation by NF1 and its role in MES glioma.

Author Response

21th November, 2021

Dear Editor, dear reviewe 1,

please find enclosed the revisions of manuscript (1456544): "Neurofibromatosis type 1 gene alterations define specific features of a subset of glioblastomas" for publication in IJMS.

We have critically revised our final manuscript point by point according to all the suggestions. Please find below all changes made in the manuscript as suggested by reviewer 1. We highlighted all changes using track changes in the revised manuscript.

Our answers:

1) It would be helpful to add more detail regarding any data out there that suggests that GBM patients (or other cancer types) with NF1 mutations respond better to immunotherapy. This is implied, but giving clinical examples would make this more impactful (referring to lines 283-295)

Response: We added a passage explaining response of NF1 mutated tumors including GBM to immunotherapy an gave examples (see track changes).

Previous passage:

… NF1 mutations lead to altered levels of cytokines, mast cells, macrophages, microglia, T and B cells, and they affect both directly immune cells and indirectly interactions between different NF1 mutated cells important for tumour microenvironment [82-84]. The finding that in NF1 associated neurofibromas and MPNST, which are derived from peripheral glia, up to 30 % of cells are macrophages, as well as a recent review of the literature led to the current definition of neurofibroma formation in NF1: Tumor initiation due to NF1 loss is followed by macrophage and mast cell recruitment which is then followed by recruitment of T and dendritic cells to enable tumour formation [85]. Half of NF1 associated low-grade gliomas were detected to harbour an immune signature, infiltrates of T cells and increased neo-antigens [15]. Therefore, the role of NF1 loss for microenvironment and tumour formation may well be adapted to the central nervous system derived malignant glial tumours, the GBM, although the literature is sparse [82]. In a recent animal model, the midkine being produced by NF1 mutant neurons activates T lymphocytes and maintains glioma growth [86].…..

New passage (new and changed sentences are underlined):

It is known that NF1 mutations correlate with high levels of leukocytes in different tumour types [81]. NF1 mutations lead to altered levels of cytokines, mast cells, macrophages, microglia, T and B cells, and they affect both di-rectly immune cells and indirectly interactions between different NF1 mutated cells important for tumour microenvironment [82-84]. The finding that in NF1 associated neurofibromas and MPNST, which are derived from peripheral glia, up to 30 % of cells are macrophages led to the current hypothesis of neurofibroma formation in NF1: Tumor initiation due to NF1 loss is followed by macrophage and mast cell recruitment which is then followed by recruitment of T and dendritic cells to enable tumour formation [85]. Half of NF1 associated low-grade gliomas were detected to harbour an immune signature, infiltrates of T cells and increased neo-antigens [15]. Therefore, the role of NF1 loss for microenvironment and tumour formation may well be adapted to the central nervous system derived malignant glial tumours, the GBM, although the literature is sparse [82]. In a recent animal model, the midkine being produced by NF1 mutant neurons activates T lymphocytes and maintains glioma growth [86].

In particular, NF1-related tumours are associated with abnormal secretion of chemokines such as C-C motif ligand (CCL) 15, CCL 2 and Macrophage colony-stimulating factor (M-CSF) leading to an increase of tumor-associated macrophages (TAM) and microglia [82,87]. Especially in GBM, loss of neurofibromin is clearly associated with the attraction of macrophages (tumor-associated macrophages, TAM) or microglia [19]. Immunotherapy strategies targeting TAM have certain potential but have only been studied in mouse models and small clinical tri-als. CCL antibodies or M-CSF receptor inhibitors reduced glioma cell invasion and resulted in longer overall sur-vival in glioblastoma mouse models [88-90].  In addition, activation of immune cell response with immune check-point inhibitors and cytokine therapy (IL-2, IFN-ß) leads to prolonged patient survival [91-95]. Ths, numerous ongo-ing clinical trials are investigating the effect of PD-1/PD-L1 antibodies in glioma. Moreover, immunotherapies seem to be not only a promising strategy for mesenchymal gliomas, but they are also an important treatment option for NF1-related melanomas, lung carcinomas or MPNST. 

2) I would also suggest adding a figure showing the role of FOSL1 both in terms of its regulation by NF1 and its role in MES glioma.

Response: Thanks to the reviewer observing that role of FOSL1 might be demonstrated more pronounced. We added an additional figure 6.

We hope that in the revised form the paper is now suitable for publication. With kind regards,

Anja Harder

Reviewer 2 Report

  1. The abstract attached to this form doesn’t match what is in the manuscript.
  2. The manuscript needs extensive English language editing and revisions. Not only for clarity and grammar, but it doesn’t seem to follow a coherent story. 
  3. Authors contradict themselves from one sentence to another. 
  4. The information in this review reads in a disjointed manner. Almost as if block of text have been put together with little thought of transitions or an overall goal of the review. 
  5. No new insights are drawn from existing data known about NF1 and gliomas or questions posed by the authors for future work. 
  6. Factually incorrect statements about NF1 syndrome and resultant reprogramming of NF1 deficient Schwann cells are made, along with incorrect conclusions of the primary literature. 

Author Response

30th November, 2021

Dear Editor, dear reviewer 2,

please find enclosed the revised manuscript (1456544): "Neurofibromatosis type 1 gene alterations define specific features of a subset of glioblastomas" for publication in IJMS.

We have critically revised our final manuscript point by point according to all the suggestions. Please find below all changes made in the manuscript. We highlighted all changes using track changes in the revised manuscript.

Grammar and language were again finally checked for mistakes (see track changes). We also used professional language revision by a native speaker.

We hope that in the revised form the paper is now suitable for publication.

With kind regards,

Anja Harder

Comments of reviewer 2 and responses

  1. The abstract attached to this form doesn’t match what is in the manuscript.

Response: Thank you for advice. We included the correct abstract.

  1. The manuscript needs extensive English language editing and revisions. Not only for clarity and grammar, but it doesn’t seem to follow a coherent story.
  2. Authors contradict themselves from one sentence to another.

Response: Thank you for advice. We performed an extensive language editing by a native speaker, and we revised the whole manuscript thoroughly to improve coherency and to delete contradiction of sentences (see track changes).

  1. The information in this review reads in a disjointed manner. Almost as if block of text have been put together with little thought of transitions or an overall goal of the review.
  2. No new insights are drawn from existing data known about NF1 and gliomas or questions posed by the authors for future work.

Response: Thank you for advice. We adapted the different text blocks to improve reading. We also added a conclusion section to summarise and to highlight the overall goal in a better way. We hope by adapting the content to outline better insights. We hope that new options (such as immunotherapy) are deduced from the conclusion section (see track changes).

  1. Factually incorrect statements about NF1 syndrome and resultant reprogramming of NF1 deficient Schwann cells are made, along with incorrect conclusions of the primary literature.

Response: We thank the reviewer for the comment and have since removed the sentence. The development and subsequent reprogramming of NF1-deficient Schwann cells has been extensively reviewed in other publications and are not the focus of this review. We focus in our review on NF1 gene alterations specifically in GBM (see track changes in chapter 1, lines 137-138).

Reviewer 3 Report

Manuscript by Scheer et al., describe role of Neurofibromatosis type 1 gene alterations in defining specific features of a subset of glioblastomas. The manuscript is very well written, cover all important aspects. 

Minor suggestion:

Authors are suggested to include the relevant and known information about what regulate the NFI and its alterations e.g. what are the upstream signaling , does NFI induce all these effects alone or some other factors co-operate or interferes with its role.

Author Response

30th November, 2021

Dear Editor, dear reviewer 3,

please find enclosed the revised manuscript (1456544): "Neurofibromatosis type 1 gene alterations define specific features of a subset of glioblastomas" for publication in IJMS.

We have critically revised our final manuscript point by point according to all the suggestions. Please find below all changes made in the manuscript. We highlighted all changes using track changes in the revised manuscript.

Grammar and language were again finally checked for mistakes (see track changes). We also used professional language revision by a native speaker.

We hope that in the revised form the paper is now suitable for publication.

With kind regards,

Anja Harder

Comments of reviewer 3 and responses

Authors are suggested to include the relevant and known information about what regulate the NFI and its alterations e. g. what are the upstream signalling, does NFI induce all these effects alone or some other factors co-operate or interferes with its role.

Response: Thank you for identifying missing information. We added another short passage (see track changes) at the end of chapter 2. Since this topic is not in the focus of the review, we hope that the explanations are suitable. If the reviewer thinks that main points are missing, we are willing to extend the new passage of the review.

New passage:

Upstream of neurofibromin mainly transmembrane receptor tyrosine kinases (RTKs) regulate extracellular ligand binding and transduce signals into the cells. They regulate signalling cascades such as the RAS/ERK pathway and therefore interfere with neurofibromin. Therapies using RTK inhibitors may therefore fail when NF1 mutations abrogate the effect on the cascade. Interestingly, Anaplastic Lymphoma Kinase (Alk) was shown to co-localize and interact with neurofibromin in Drosophila and was demonstrated to activate neurofibromin regulated RAS signalling in the nervous system [61]. A direct interacting partner of neurofibromin is also the membrane bound late endosomal/lysosomal adaptor and MAPK and mTOR1 activator (LAMTOR), a negative regulator of the mTOR pathway. [62,63] Other interacting partners, although very important such as Sprouty-related, EVH1 domain-containing protein 1 (SPRED1) that recruit neurofibromin from the cytosol to facilitate the transport to the plasma membrane, will not discussed here in detail.  

Round 2

Reviewer 1 Report

Changes made were found to be acceptable. The additional Figure 6 was a nice addition

Author Response

Dear reviewer 1,

thank you very much for your decision that the manuscript is suitable now.

Thank you for your advice.

Yours sincerely,

Anja Harder

Reviewer 2 Report

The authors have made extensive edits and revisions to this review. The graphics and figures are well made. I thank the authors for their efforts. 

There are still significant issues with the presentation. It is not clear to this reviewer the point(s) the authors are trying to make with this manuscript. Even the added conclusion section offers no real summary or discussion based on the data presented. It is just a basic list of the some bullet point style statements. 

This review adds little to those already published on the topic, is poorly organized, and offers no new conclusions. Nor does it postulate new questions or suggest future directions for the field. If this be expanded in your conclusion section, readers would find the information useful and the work would be dramatically improved. 

While the English writing and organization has been massively improved from the last version, certain sections still remain unclear and contradictory. 

Some specific comments:

Lines 169-170: As written the authors are suggesting that neurofibromas express NF1 and that is spliced the same as it is within normal Schwann cells. Since the tumor cells are Schwann cells, this is technically correct. Although, be definition the Schwann cells driving formation of the benign neurofibroma have biallelic loss of function mutations in NF1, so they are not expressing the gene product (neurofibromin). There are of course other Schwann cells within the tumor that are heterozygous for the NF1 mutation. They authors should clarify when they are saying “expression profiles” in this section as to if they are referring to splicing, mRNA expression, or protein expression. It is not at all clear the message they are trying to get across.

Lines 174-180: The authors should add which splice variants are expressed in various neuronal tissue. This is known and relates directly to the work they are reviewing. Main splice variant expressed for Neurons and glial cells of the CNS and PNS should be listed.

Line 228-229: “In fact, very few genotype-phenotype correlations exist, such as a higher and more aggressive tumour load in patients with microdeletions [66].” This statement is not clear and doesn’t make sense as written. Maybe something like: In fact, very few genotype-phenotype correlations exist, with the exception of higher and more aggressive tumour load in patients with microdeletions.

Line 260-261: Patients with NF1 do not have biallelic loss of function mutations in NF1. This would of course be embryonically lethal. They can develop Schwann cells tumors in which a rare Schwann cell loses function of the second copy of NF1. I think the authors are aware of this, but the wording should be revised to reflect as such.

Lines 265-266: Authors state “the development of NF1 and the subsequent reprogramming of Schwann cells have been extensively reviewed and are not the focus of this review. “ They have a whole section dedicated to NF1 syndrome. They should at the very least cite some of the reviews which they refer to in this statement.

Lines 438-439: Authors state: “We also demonstrated that glioma cell invasion is specifically regulated via another domain, the LRD of neurofibromin.” How have the authors demonstrated this? There is no new research data presented in this review. If they have shown this in published research work, they should cite the work.

Lines 439-440: “Nevertheless, also other domains such as GRD, Sec14-PH domain, pre-GRD and CTD have been described…” This comes immediately after a sentence discussing the GRD in another context. It doesn’t make sense as written.

Author Response

14thth December, 2021

Dear Editor,

please find enclosed the newly revised manuscript (1456544): "Neurofibromatosis type 1 gene alterations define specific features of a subset of glioblastomas" for publication in IJMS.

We have critically revised everything according to reviewer’s suggestions. Please find below all changes made in the manuscript. We highlighted all changes using track changes. Grammar and language were again finally checked for mistakes.

We hope that in the secondly revised form the paper is now suitable for publication.

With kind regards,

Anja Harder

Comments of reviewer 2 and responses

  1. The authors have made extensive edits and revisions to this review. The graphics and figures are well made. I thank the authors for their efforts. 

  1. There are still significant issues with the presentation. It is not clear to this reviewer the point(s) the authors are trying to make with this manuscript. Even the added conclusion section offers no real summary or discussion based on the data presented. It is just a basic list of the some bullet point style statements. 

Response: Thank you for the comment. We have completely rewritten the conclusion to include a summary of the review and future directions (see lines 403-432).

New passage:

Despite advances in surgery and molecular therapeutics, the prognosis for patients with GBM remains dismal. The highly infiltrative and heterogenous nature of the tumour is rendering standard therapeutic strategies ineffective. NF1 is one of the driver genes for MES GBM. In this review, we discussed the molecular characteristics of MES GBM, NF1 gene mutation, and dysregulation in NF1-associated and non-NF1 associated cancers, particularly GBM. However, many questions remain unanswered. MES GBM gene expression is influenced by dysregulated neurofibromin signalling and tumour microenvironment [125]. In NF1-null or silenced MES GBM, the microenvironment is heterogenous with hypoxic core and perivascular niche each secreting different cytokines and chemokines that drive tumor malignancy. Given the complexity of the bi-directional interaction, the design of therapeutics must take into consideration the dynamic crosstalk among the various players such as glioma cells, immune cells (immunosuppressive versus pro-inflammatory), endothelial cells, among others. Macrophages and microglia cells secrete factors that promote tumour growth. Are we able to re-educate these cells in the NF1-null microenvironment to achieve the anti-tumour function?  Studies conducted by Pyonteck et al. using a brain-penetrant inhibitor of colony-stimulating factor 1 receptor (CSF-1R) showed a significant decrease of pro-tumorigenic tumor-associated macrophages [100], suggesting that blocking CSF-1R signalling may re-educate the immunosuppressive macrophage to pro-inflammatory cells. Another CSF-1R tyrosine kinase inhibitor, PLX3397, prevented the differentiation of monocytes into immunosuppressive macrophages [126]. Unfortunately, PLX3397 was ineffective in a phase II trial in treating recurrent GBM [101]. Thus, understanding the intricate relationship between these cells and their associated gene expression changes may help develop more effective immunotherapeutics. Given that GBM subtypes are not static, it is evident that multiprong therapy may afford a better therapeutic outcome. Previous publications have shown that CEBP-b, STAT3, NF-kB, FOSL2 are some of the transcription factors (TFs) that play a role in NF1-loss-associated MES transition [127]. Among these TFs, STAT3 and CEBP-b have been shown to associate with the hypoxic microenvironment [29,128], enriched with immunosuppressive tumor-associated macrophages [129]. Gabrusiewicz and colleagues showed that GBM-derived exosomes triggered the release of STAT3 in monocytes and led to upregulation of programmed death-ligand 1 (PD-L1) and a shift to the immunosuppressive phenotype [130]. Several STAT3 inhibitors are currently in clinical trials. These inhibitors were designed to be used concurrently with conventional radiation (NCT03514069) and chemotherapy (NCT02315534). Other inhibitors that target the molecules in the STAT3 pathway, such as JAK1/JAK2, are also being evaluated in phase I trial for patients with newly diagnosed GBM (NCT03514069). While we await the result from these trials, identifying other NF1-loss associated master regulators and their inhibitors may improve the treatment options for patients with MES GBM.

.

  1. This review adds little to those already published on the topic, is poorly organized, and offers no new conclusions. Nor does it postulate new questions or suggest future directions for the field. If this be expanded in your conclusion section, readers would find the information useful and the work would be dramatically improved. 

Response: Thank you for the advice. As commented under 2), we have completely rewritten the whole conclusion section. Therefore, please refer in detail to point 2). We hope that by the new arrangement the topic is better understood.

  1. While the English writing and organization has been massively improved from the last version, certain sections remain still unclear and contradictory. 

Response: We again revised the whole manuscript and made some corrections; we hope that the manuscript is now clear and suitable to read.

  1. Some specific comments:

Lines 169-170: As written the authors are suggesting that neurofibromas express NF1 and that is spliced the same as it is within normal Schwann cells. Since the tumor cells are Schwann cells, this is technically correct. Although, be definition the Schwann cells driving formation of the benign neurofibroma have biallelic loss of function mutations in NF1, so they are not expressing the gene product (neurofibromin). There are of course other Schwann cells within the tumour that are heterozygous for the NF1 mutation. They authors should clarify when they are saying “expression profiles” in this section as to if they are referring to splicing, mRNA expression, or protein expression. It is not at all clear the message they are trying to get across.

Response: Thank you for the advice, we reworded the sentence to clarify that the expression profiles refer to spliced RNA profiles (see insertion in bold, see track changes lines 165-166).

Previous version:

… It was also shown that benign tumours and peripheral nerves share the same expression profile, indicating that in benign tumours, NF1 may be spliced identically…

New version:

It was also shown that benign tumours and peripheral nerves share the same spliced RNA expression profile, indicating that in benign tumours, NF1 may be spliced identically.

Lines 174-180: The authors should add which splice variants are expressed in various neuronal tissue. This is known and relates directly to the work they are reviewing. Main splice variant expressed for Neurons and glial cells of the CNS and PNS should be listed.

Response: Thank you for the comment. We rewrote the passage /see below) to be clear about the splice variants (see also track changes lines 156-185). We also added a more detailed description concerning the splice variants. If the reviewer wishes to have included much more information, we can add a supplementary file (please find a suggestion at the end).

New passage:

Many studies indicate the importance of NF1 splice variants, of which five are analyzed on an experimental level [37-40]. In general, the gene product neurofibromin isoform type 2 (NP_000258.1, 2818 amino acids (aa)) is expressed ubiquitously and shows a 10 times higher Rat sarcoma GTPase activating protein (Ras-GAP) activity than isoform 1 (NP_001035957.1; 2839aa) [41]. It is preferentially expressed in differentiated cells [37,42]. Isoform 1 contains 21 additional amino acids encoding for the alternatively spliced exon 23a. The alternatively spliced exon 23a (exon 31 according to the new nomenclature) [43,44] is placed midst of the GTPase-activating domain (GAP) related domain (GRD). The Ras-GAP activity therefore, depends on 23a exon splicing. Isoform 1  represents the most abundant isoform [45]) and is expressed in adult tissues of neural crest lineage [46]. Still, there is evidence for a tissue-specific accumulation of splice variants, co-existence of different splice variants in the same cell type and a correlation between the protein expression level and tissue type [39,47,48]. It was also shown that benign tumours and peripheral nerves share the same spliced RNA expression profile, indicating that in benign tumours, NF1 may be spliced identically. In the CNS, NF1 isoform 2 is preferentially expressed in pure glial cultures; while isoform 1 is predominantly expressed in neuronal cells [49]. Among the different splice variants, the National Center for Biotechnology Information (NCBI) reference sequence NM_000267.3 is most widely used for variant analysis. Accumulation and expression of splice variants are specific to developmental stage and tissue [50]: the splice-variant resulting from alternative splicing of exon 9a adds ten amino acids to the protein sequence and is mainly located in the central nervous system. Studies in mice showed increased expression levels during the first postnatal week suggesting a role for maturation and differentiation of neurons [39,51,52]. The alternative spliced exon 10a-2 is located between exon 10a and 10b and adds fithteen additional amino acids. The resulting additional motive forms a transmembrane segment that does not appear in other variants. Although expression was detected in every human tissue pointing to a housekeeping function [40]. Alternative splicing of exon 48a results in additional eighteen amino acids and is discussed to play a role in differentiation of fetal and adult cardiac and skeletal muscle [38,53,54] Interestingly, alternative spliced exons 29 and 30 lead to three different protein isoforms: ex29-, ex30- and ex29-/30- [55]. Except for ex29- which is only apparent in the brain, these variants are ubiquitously expressed, but no variant-specific function has been described so far.

Line 228-229: “In fact, very few genotype-phenotype correlations exist, such as a higher and more aggressive tumour load in patients with microdeletions [66].” This statement is not clear and doesn’t make sense as written. Maybe something like: In fact, very few genotype-phenotype correlations exist, with the exception of higher and more aggressive tumour load in patients with microdeletions.

Response: We included the more precise suggestion of the reviewer (see track changes, lines 232-233).

Line 260-261: Patients with NF1 do not have biallelic loss of function mutations in NF1. This would of course be embryonically lethal. They can develop Schwann cells tumors in which a rare Schwann cell loses function of the second copy of NF1. I think the authors are aware of this, but the wording should be revised to reflect as such.

Response: Thanks for the advice, we improved wording (see track changes, lines 262-265).

Lines 265-266: Authors state “the development of NF1 and the subsequent reprogramming of Schwann cells have been extensively reviewed and are not the focus of this review. “ They have a whole section dedicated to NF1 syndrome. They should at the very least cite some of the reviews which they refer to in this statement.

Response:  According to the reviewer’s suggestion we added new references (75, 86-91) to underline the statement (see line 267 for track changes).

Lines 438-439: Authors state: “We also demonstrated that glioma cell invasion is specifically regulated via another domain, the LRD of neurofibromin.” How have the authors demonstrated this? There is no new research data presented in this review. If they have shown this in published research work, they should cite the work.

Response: Thank you for the comment. We have rewritten the conclusion and this sentence is deleted from the manuscript.

Lines 439-440: “Nevertheless, also other domains such as GRD, Sec14-PH domain, pre-GRD and CTD have been described…” This comes immediately after a sentence discussing the GRD in another context. It doesn’t make sense as written.

Response: Thank you for the comment. Since we have rewritten the conclusion section this sentence is also deleted from the manuscript.  

Suggestion for a supplementary file

Type of neuro-fibromin

Exon insertion/deletion old nomen-clature

Exon insertion/ deletion                 new nomen-clature

Amino acids

Expressed in

Function

Reference

Isoform I (Type 1)

-23a exon

2818 aa

neurons, brain (cere-bral cortex, brainstem, cerebellum)

ratio of iso-forms is im-portant for tissue de-velopment

Gutman et al., 1995

Isoform II (Type 2)

30alt31_1-63 (1-21)

+23a exon, 63nt

exons 30-31, 30alt31

2839, 21 aa in GRD do-main

glial cells, adrenal glands, ovaries

23a is re-gulatory for  Ras-GAP activity, (reducing)

Gutman et al., 1995

Type 3

56alt57_1-54 (1-18)

+48a (54nt)

exons 56-57, 56alt57

2836, 18 aa at C-ter-minal end

muscles,  heart, high-ly ex-pressed in embryos

muscle develop-ment and differen-tiation

Gutmann et al., 1995

Type 4

30alt31_1-63 (1-21)

56alt57_1-54 (1-18)

+23a, + 48a (63nt+54nt)

exons 30-31, 30alt31;

exons 56-57, 56alt57

2857aa, 21 aa in GRD,  18 aa in CTD

heart and muscles

Gutmann et al., 1995

11alt12_1-30 (1-10)

+9a exon (30nt) after nt 1260 in cDNA

exons 11-12, 11alt12

2828, 10aa in N-terminal region

CNS neu-rons, en-riched in forebrain, postmitotic neurons

maturation and differen-tiation of neurons

Danglot et al., 1995; Geist and Gutman 1996

12alt13_1-45 (1-15)

+10a-2 (45nt) between exons 10a and 10b

exons12-13, 12alt13

2833 aa,   15 aa in N-terminal region

ubiquitously expressed but at low levels

Transmem-brane seg-ment that is absent in other iso-forms, spe-cific locali-saton to perinuclear granular structures

Kaufmann et al., 2002

NF1 delta E43

NF1 deltaE51

-exon 43 (7553-7675nt)

-exon 51 (7553-7675nt)

2777aa,

delta 2518-2559aa,

no NLS

lung, liver, placenta, kidneys, and skeletal muscle of adult humans

no nuclear localization signal, low expression in neurons suggesting  importance in neural nuclei

Vanden-broucke et al., 2002 and 2004; 

Gutmann et al., 1995

Reviewer 3 Report

The authors have addressed the raised concerns.

The manuscript can be accepted in the current form.

Author Response

Dear reviewer 3,

thank you very much for your decision that the manuscript is suitable now.

Thank you for your advice.

Yours sincerely,

Anja Harder

Round 3

Reviewer 2 Report

The authors have addressed the major issues I had with the review. Thank you for the extensive revisions. The content and flow/presentation of the work is now much better and the factually incorrect or highly misleading statements have been fixed.